# Raloxifene Suppresses Tumor Growth and Metastasis in an Orthotopic Model of Castration-Resistant Prostate Cancer

**DOI:** 10.3390/biomedicines10040853

**Published:** 2022-04-05

**Authors:** Hannah Palmer, Mhairi Nimick, Aloran Mazumder, Sebastien Taurin, Zohaib Rana, Rhonda J. Rosengren

**Affiliations:** 1Department of Pharmacology and Toxicology, University of Otago, Dunedin 9016, New Zealand; hannahrosepalmer91@gmail.com (H.P.); mhairi.nimick@otago.ac.nz (M.N.); aloran.mazumder@gmail.com (A.M.); sebastient@agu.edu.bh (S.T.); 2Department of Biochemistry, University of Otago, Dunedin 9016, New Zealand; zohaib.rana@otago.ac.nz

**Keywords:** CRPC, AR-prostate cancer, raloxifene, curcumin derivative, combination therapy, anti-cancer agent, orthotopic model

## Abstract

Androgen receptor (AR)-castrate-resistant prostate cancer (CRPC) is an aggressive form of prostate cancer that does not have clinically approved targeted treatment options. To this end, the cytotoxic potential of raloxifene and the synthetic curcumin derivative 2,6-bis (pyridin-4-ylmethylene)-cyclohexanone (RL91) was examined in AR-(PC3 and DU145) cells and AR+ (LnCaP) CRPC cells. The results showed that both raloxifene and RL91 elicited significant cytotoxicity across three cell lines with the lowest EC_50_ values in PC3 cells. Additionally, the two drugs were synergistically cytotoxic toward the PC3, DU-145 and LNCaP cell lines. To determine the effect of the drug combination in vivo, an orthotopic model of CRPC was used. Male mice were injected with PC3 prostate cancer cells and then treated with vehicle (5 mL/kg), raloxifene (8.5 mg/kg, po), RL91 (8.5 mg/kg, po) or a combination of raloxifene and RL91 for six weeks. Sham animals were subjected to the surgical procedure but were not implanted with PC3 cells. The results showed that raloxifene decreased tumor size and weight as well as metastasis to renal lymph nodes. However, combination treatment reversed the efficacy of raloxifene as tumor volume and metastasis returned to control levels. The results suggest that raloxifene has tumor suppressive and anti-metastatic effects and has potential for further clinical use in AR-CRPC.

## 1. Introduction

Prostate cancer is amongst the most commonly diagnosed malignancies and a leading cause of cancer death in men. Globally, new cases of prostate cancer are estimated to reach approximately 1.7 million by 2030 [1]. Prostate cancer is initially dependent on androgens for growth, and thus androgen withdrawal therapy (often via castration) can reduce the growth of the tumor. However, over time, the cancer can progress to an androgen-independent form classified as castrate-resistant prostate cancer (CRPC), where tumors can grow in the absence of androgens [2,3,4]. As tumors still express androgen receptors (AR), AR antagonists, such as enzalutamide, are administered in patients. However, an androgen-independent tumor differentiates further, triggered by activation of the PI3K/Akt pathway into an AR-phenotype in 11–24% of patients, rendering the AR antagonist treatment ineffective [5,6,7,8]. AR-prostate tumors have a poor prognosis, with the majority of patients dying within two years [9]. Unfortunately, no treatment options are available clinically. One phase I trial using carboplatin and etoposide was conducted in patients with AR-tumors; however, an objective response was only observed in four out of 56 patients, with an overall survival of 9.6 months [10]. Therefore, treatments urgently need to be identified for AR-CRPC prostate cancers.

Evidence suggests that the expression of estrogen receptor (ER)-α and ER-β are modulated in prostate cancer and function independently of the AR. ER-α functions as an oncogene, enhancing cell survival and proliferation by mediating PI3K/Akt and RAS/MAPK signaling pathways, [11]. Bonkhoff et al. (1999) reported that ER-α expression was the highest in metastatic prostate cancer lesions in the bone and lymph node [11]. ER-α expression also significantly correlated with high Gleason scores (>4) [11]. In contrast, ER-β functions as a tumor suppressor in prostate cancer, with partial knockdown observed in prostatic intraepithelial neoplasia [12]. Therefore, ER-α antagonists and ER-β agonists might be useful for the treatment of prostate cancer. For this reason, selective estrogen receptor modulators (SERMs), such as raloxifene, have been tested in AR-(PC3, DU-145) and AR+ (LnCaP) prostate cancer cells inducing apoptosis at micromolar concentrations [13]. As enhanced PI3K/Akt signaling is observed in prostate cancer (LnCaP, PC3 and DU-145) cells, a PI3K/Akt signaling inhibitor could potentially enhance the efficacy of SERMs. We have previously shown that RL91 inhibited the PI3K/Akt signaling pathway by 98% in breast cancer cells. Therefore, in this study, raloxifene has been examined in combination with a synthetic curcumin analog, RL91, with the aim of increasing efficacy. These drugs were examined for in vitro potency and this was followed by in vivo efficacy in an orthotopic model of CRPC in mice.

## 2. Materials and Methods

### 2.1. Cell Maintenance

All three cell lines were purchased from American Type Culture Collection (Manassas, VA, USA). All cell culture and general reagents and chemicals were purchased from Sigma-Aldrich (St. Louis, MO, USA), unless otherwise specified1.8. Cells were maintained in DMEM media supplemented with 10% FBS, 100 U/mL penicillin, 100 Ug/mL streptomycin and 2.2 g/L NaHCO_3_. Cells were cultured in 75 cm^2^ flasks and incubated in 5% CO_2_/95% humidified air and at 37 °C.

### 2.2. Cytotoxicity Assays

PC3 (4 × 10^3^ cells/well), DU-145 (6 × 10^3^ cells/well) and LNCaP (4 × 10^3^ cells/well) cells were seeded in 96 well plates in 150 µL DMEM/HamF12 supplemented with 5% FBS, 100 U/mL penicillin, 100 ug/mL streptomycin, 2.2 g/L NaHCO_3_ and incubated for 24 h at 37 °C. Cells were treated with ranging concentrations of raloxifene and RL91 for 72 h. Vehicle control cells were treated with DMSO (0.1%). Cell number was determined using the sulforhodamine B assay [14]. The concentration of each compound required to decrease the cell number by 50% of the vehicle control (EC_50_) was determined by non-linear regression using Prism software. EC_50_ values are provided as the mean ± SEM from three independent experiments performed in triplicate. Drug combination studies were then performed in PC3, LNCaP and DU-145 cells which were seeded as above and treated with raloxifene (2–10 µM), RL91 (1–3 µM) or a combination of the two for 72 h. Higher concentrations of RL91 and raloxifene were required in LNCaP cells, as they were less responsive to drug treatment. Results are expressed as cell number (% of control) and were obtained from three independent experiments performed in triplicate. Synergism was determined by the method described by Chou and Talalay [15] using CompuSyn 1.0 software (Paramus, NJ, USA) [16].

### 2.3. Cell Cycle Distribution

Flow cytometry was used to analyse DNA content in order to determine cell cycle phases. PC3 cells (1 × 10^5^/well) were plated in 6-well plates and treated with raloxifene (10 µM), RL91 (2 µM), a combination of both drugs, or vehicle control (0.1% DMSO) and incubated for 24–48 h. Cells were then harvested and centrifuged at 1000× *g* for 5 min. The pellet was re-suspended in PBS and centrifuged two more times. The final pellet was re-suspended in 300 μL PBS. 600 μL of ice-cold ethanol was then added in a dropwise manner to fix the cells. Cell suspensions were kept at 4 °C for 1 h. Tubes were centrifuged and the pellet was then washed in PBS, centrifuged, the supernatant discarded and the cell pellet re-suspended in 300 μL PBS supplemented with 0.1% glucose. Each tube was then heated at 37 °C for 5 min. 0.5 μL of RNase A (20 mg/mL) (Invitrogen, Auckland, New Zealand) was added and tubes were incubated for 5 min before being returned to ice. 15 μL of propidium iodide (PI) (1 mg/mL) (Sigma-Aldrich, St. Louis, MO, USA) was added and tubes were left to incubate in the dark for 1 h at 4 °C. The samples were then analysed via flow cytometry using a FACSCaliber machine where PI was detected in the FL-2 channel. Data were acquired and analyzed using CellQuest Pro 1.0 software (BD Biosciences, San Jose, CA, USA). Results are expressed as the number of gated cells in each phase of the cell cycle as a percentage of the total number of cells and were obtained from three independent experiments performed in triplicate.

### 2.4. Apoptosis Analysis

Apoptosis was determined through the externalisation of phosphatidylserine on the extracellular membrane as previously described [17]. PC3 cells were seeded at a density of 1 × 10^5^/well in 6-well plates. Cells were treated with raloxifene (10 µM), RL91 (2 µM), a combination of both at the same concentrations, or a vehicle control (0.1% DMSO), and incubated for 24–72 h. At the end of the treatment period, the cells were harvested and then centrifuged at 1000× *g* for 5 min. The pellet was re-suspended in PBS and centrifuged again. The final pellet was re-suspended in 100 μL of FACS binding buffer (10 mM HEPES/NaOH, pH 7.4, 140 mM NaCl, 5 mM CaCl2), Annexin V (0.5 μL) (Roche Diagnostics, Mannheim, Germany) and PI (1 μL). The samples were incubated in the dark for 15 min and then analyzed on a FACSCaliber machine. Annexin-V FLOUS and PI were detected in the FL-1 and FL-2 channels, respectively. Data were acquired and analyzed using CellQuest Pro software. Results are expressed as the number of apoptotic cells as a percentage of the total number of cells and were obtained from three independent experiments performed in triplicate.

### 2.5. Orthotopic Model of Castration Resistant Prostate Cancer

Male SCID mice (9 weeks old) were purchased from Animal Resources Centre (WA, Australia). All procedures were approved by the University of Otago Animal Ethics Committee (protocol number 8/14) and all procedures were performed in accordance to the protocols outlined in protocol number 8/14. Mice were housed in pathogen-free conditions with sterile woodchip bedding with access to food (Reliance rodent diet) and water ad libitum. Mice were housed in a 21–24 °C environment on a scheduled 12 h light/dark cycle. Mice were anesthetized with inhaled isoflurane (4%) and once reflex to toe pinch was absent, mice were administered analgesic (carprofen at 5 mg/kg, sc) before a small skin incision was made approximately 1 cm above the penis. This was followed by an incision into the abdominal wall. Using forceps, the bladder was externalized and lifted to visualize the ventral prostate lobes. A Hamilton syringe was used to inoculate 2 × 10^5^ of PC3 cells in 20 uL of DMEM serum-free media directly into the proximal area of the ventral prostate lobe. Sham control mice received 20 uL of DMEM serum-free media only. Absorbable sutures were then used to close both the abdominal wall and skin incisions. Three weeks following surgery, mice were randomly assigned to one of four treatment groups with four to five animals per group. Mice were orally gavaged daily for 42 days with vehicle control (1:1 peanut oil:water 5 mL/kg), RL91 (8.5 mg/kg in peanut oil), raloxifene (8.5 mg/kg in water) or a combination of RL91 and raloxifene.

### 2.6. Tissue and Blood Collection

Following six weeks of treatment, mice were euthanized by CO_2_ inhalation. Blood was collected from the inferior vena cava using a 20-gauge needle attached to a heparin-coated syringe and plasma was separated and stored at −20 °C. Prostate glands and all visible lymph nodes in the surrounding area were collected from all animals. Major organs were harvested and weighed. Prostate volume was determined with digital calipers and the lymph node area was determined using Image J 1.8 software (Bethesda, MD, USA). All collected tissue was weighed and stored in 4% paraformaldehyde overnight at 4 °C. The paraformaldehyde was replaced with a 30% sucrose solution and the tissues were stored for a further 24 h before being frozen in optimal cutting temperature compound (OCT), on an aluminum block submerged in liquid nitrogen, and stored at −20 °C until further processing.

### 2.7. Immunohistochemistry

Lymph nodes, lungs and tumors were sliced (10 µm) in a coronal plane using a cryostat (Leica). Slices were mounted onto poly-L- lysine coated microscope slides and left to dry overnight before being frozen at −20 °C until required. Slices were thawed for 30 min and then washed 2 times in PBS for 5 min. Tissues were fixed with 100% acetone for 10 min at room temperature. Endogenous peroxidases were blocked with 0.3% H_2_O_2_ in methanol for 20 min at room temperature and slides were then rinsed again in PBS for 2 min. Antigen retrieval was carried out for all antibodies except CD105 by immersing slides in a citrate buffer (10 mM citric acid, 0.05% Tween-20, pH 6.0) for 10 min at 90–95 °C. Once cooled, slides were again rinsed in PBS. 1% Triton X-100 was applied for 30 min for permeabilization of cell membrane. Slides were then blocked for 1 h at room temperature with 1.5% goat serum blocking solution (in PBS). An Avidin/biotin blocking kit for all antibodies (i.e., CD105, Ki67, ERα and ERβ) and the mouse on mouse (M.O.M.) immunodetection kit, anti-mitochondria antibody only, (Vector Laboratories, Burlingame, CA, USA) were then used according to manufacturer’s instructions. After washing, sections were incubated with Steptavidin-HRP for 30 min and then stained with DAB for 10–20 min in the dark and then washed with ddH_2_O. An ApopTag Peroxidase kit (Millipore, Burlington, MA, USA) was used as per the manufacturer’s instructions. The tissue was then counterstained with hematoxylin QS, dehydrated in ethanol, soaked in xylene and mounted with DPX. Sections were scanned using the Aperio 12.3.3 software (Leica, Mt Waverly, Australia) at 20× magnification. Data was then analysed using Positive Pixel Count v9, Nuclear v9 or Microvessel Analysis v1 algorithms in a 4× magnified section. Images presented are a 20× magnification and are a representative photo from either four or five tumors per treatment group or represent all lymph nodes collected. Each of these was further analysed into three sections per individual tissue.

### 2.8. Hematoxylin and Eosin Staining

Tissue slices were immediately fixed in 70% ethanol for 1 min and rinsed with ddH_2_O for 5 min. Hematoxylin was applied briefly for 10 sec before slides were immersed in tap water and thoroughly rinsed for 20 sec. Slices were then rinsed in 70% ethanol for 1 min before a counterstain of eosin was applied for 10 sec. Slides were then immersed in 95% ethanol and thoroughly rinsed for 20 sec before a second 95% ethanol rinse. Slides were rinsed in xylene for 10 min before being mounted with DPX and a coverslip applied. Sections were scanned using the Aperio 12.3.3 software (Leica, Mt Waverly, Australia)) using 20× objective magnification.

### 2.9. Statistical Analysis

Data are expressed as mean ± SEM. All data with one factor was analysed with a one-way ANOVA and Bonferroni post-hoc test. When time was an additional factor, data was analysed with a two-way ANOVA and Bonferroni post-hoc test. Immunohistochemistry in the tumor sections was analyzed with an unpaired *t*-test. All data with unequal variance were log transformed and then reanalyzed with the appropriate statistical test. For all analysis, *p* < 0.05 was the minimum requirement for a statistically significant difference.

The datasets generated during and/or analyzed during the current study are available from the corresponding author on reasonable request.

## 3. Results

The cytotoxic potential of raloxifene and RL91 was examined in PC3, LNCaP and DU-145 cells. The results showed that RL91 elicited significant cytotoxicity across the three cell lines with the lowest EC_50_ reported in PC3 cells (1.9 ± 0.1 μM). Similarly, raloxifene-mediated cytotoxicity was more pronounced in PC3 cells with an EC_50_ of 11.6 ± 0.5 μM (Table 1).

A combination of the two compounds was then examined in PC3, DU-145 and LNCaP cells. The results showed that the two drugs were synergistically cytotoxic toward all three cell lines, as RL91 (2 µM) and raloxifene (10 µM) elicited combination indexes of 0.81, 0.73 and 0.91 in PC3, DU-145 and LNCaP cells, respectively. Furthermore, the drug combination was again more potent in PC3 cells compared to DU-145 and LNCaP cells (Figure 1). Specifically, the combination of RL91 (2 μM) and raloxifene (10 μM) reduced cell number by 95% of control. A wider range of concentrations of each drug individually and in combination was also examined in PC3 and LNCaP cells (i.e., the most sensitive and most resistant cell lines). The results showed that PC3 cells were more responsive to combination treatment and isobologram analysis showed persistent synergism at a wide range of concentrations (Appendix A).

Since the combination treatment was most effective in PC3 cells, further analysis was only performed with this cell line. Specifically, combination treatment for 24 h caused a 1.5-fold increase in the number of cells in the G2/M phase of the cell cycle (38.3 ± 0.9 vs. 25.4 ± 0.5% for combination and control, respectively, Figure 2A) and this effect was also greater than that elicited by RL91 as a single treatment. Raloxifene has been reported to arrest cells in the G1 phase of the cell cycle [18] and this was also observed in this study when raloxifene was used as a single treatment (66.5 ± 0.5% of cells in the G1 phase for raloxifene vs. 51.9 ± 0.7% for control). However, G2/M phase arrest dominated following combination treatment and the effects were sustained over 48 h. The arresting of cells in the G2/M phase of the cell cycle was a driver of sustained apoptosis, as the number of cells undergoing apoptosis was increased following either RL91 or combination treatment for 48 h (Figure 2B). By 72 h of combination treatment, 28% of PC3 cells were apoptotic and this effect was significantly greater than that elicited by any other treatment (Figure 2B).

Since the combination of raloxifene + RL91 showed synergistic cytotoxicity as well as increased G2/M phase arrest and induced apoptosis, the ability of the drug combination to modulate tumor growth in an orthotopic model of CRPC was conducted. Male mice bearing orthotopic prostate tumors were administered a daily oral dose of vehicle, RL91 (8.5 mg/kg), raloxifene (8.5 mg/kg), or a combination of the two for six weeks. The results showed that raloxifene treatment significantly reduced tumor volume by 70% compared to vehicle (529 ± 204 vs. 1784 ± 503 mm^3^, for raloxifene and vehicle control, respectively, *p* < 0.05), while RL91 decreased tumor volume 61% compared to vehicle control (Figure 3A), which was not significantly different. Surprisingly, the addition of RL91 reversed the efficacy of raloxifene. Following six weeks of treatment, RL91 in combination with raloxifene elicited only a 42% reduction in tumor volume compared to vehicle control (Figure 3A). Furthermore, all prostate volumes were larger than that of normal prostate glands from the sham treatment group (70 ± 15 mm^3^, Figure 3B). No systemic toxicity was observed as body weight gain, organ weight, plasma creatinine and ALT activity all remained within the normal range (Table 2). Furthermore, food and water consumption as well as mouse behavior was the same in all treatment groups.

Since the combination of RL91 + raloxifene was ineffective, we concentrated on the mechanism of action for raloxifene as a single drug. Thus, immunohistochemical analyses of tumor sections were performed on tumors from mice treated with the vehicle control or raloxifene. The results showed that raloxifene treatment decreased microvesssels in the tumor 38% compared to control, as shown by CD105 staining, but this effect was not statistically significant (Figure 4A). The number of proliferating cells, as determined by Ki67 staining, was also decreased 42% compared to control and this was also not statistically different (Figure 4B). However, the number of apoptotic cells was significantly increased to 361% above control, as shown by analysis of ApopTag staining (Figure 4C). Raloxifene also significantly decreased the expression of both ERα (decreased 84% compared to control, Figure 4D) and ERβ (decreased 92% compared to control, Figure 4E) within the tumor.

To determine whether any of the drug treatments altered metastasis of the PC3 cells, the lymph nodes and lungs from all experimental animals were examined. Similar to the results from the primary tumors, the combination treatment of RL91 and raloxifene did not alter metastasis, as determined by measuring lymph node size (Figure 5). While raloxifene decreased the lymph node size when all lymph nodes were grouped together, this effect was not significantly different from vehicle control (Figure 5A). However, when lymph nodes were analyzed by their regional location, raloxifene treatment decreased the size of renal lymph nodes 62% compared to vehicle treated mice (4.5 ± 0.6 vs. 11.6 ± 2.26 mm^2^, respectively) (Figure 5B).

While increased lymph node size is an important indicator of cancer cell invasion, it was necessary to confirm that the enlarged area was due to infiltration of the inoculated PC3 cells. Immunohistochemistry using human mitochondria primary antibody was conducted on sections from all lymph nodes collected. Thus, the mitochondria of human PC3 cells were positively stained and nuclei were counterstained by hematoxylin. This resulted in a clear contrast between human metastatic cells (brown) and mouse lymph tissue (blue) (Figure 6A). Images were then analysed using ImageScope software. The results of the imaging analysis showed that raloxifene significantly decreased the percentage of mitochondrial staining by 60% compared to vehicle control (22.4 ± 3.4 vs. 56.9 ± 6.1%, for raloxifene and vehicle control treatment, respectively) (Figure 6B) and the addition of RL91 abolished this effect. In this model, there was also no evidence of metastasis to the lungs in any of the treatment groups as confirmed by a registered pathologist. Taken together the in vivo results showed that RL91 + raloxifene reversed the tumor and metastasis suppression effects elicited by raloxifene.

## 4. Discussion

AR-prostate cancer has a poor prognosis, with the majority of patients dying within two years and no clinically approved targeted treatment options [9,19]. Thus, there is a need to discover treatments that target prostate cancer cells independently of the AR. Previous studies have shown that ER and PI3K/Akt signaling can drive prostate cancer growth independently of the AR, and this study targeted both by using the combination of raloxifene and RL91 [7,8,11]. The benefit of using combination treatments is that it can lower the doses of individual drugs, and thus can lower toxicity and the chances of incurring resistance [20,21]. Raloxifene as an individual drug and when used in combination with RL91 had a consistently greater cytotoxic effect toward PC3 cells as compared to both DU-145 and LnCaP cells. The higher potency of raloxifene in PC3 cells may be attributed to the higher expression of ER-α and ER-β, whereas DU-145 and LnCaP lack ER-α expression but express only ER-β [22]. However, as raloxifene is an example of a SERM, its effects as an agonist/antagonist have not yet been studied in the prostate. Previous studies have proposed that raloxifene can act via both ER-isoforms, inhibiting signaling pathways such as RAS/MAPK and PI3K/Akt that induce cell cycle arrest and induce apoptosis in various forms of cancers, including breast and prostate cancer [23,24].

As PI3K/Akt signaling is highly active in AR-prostate cancer, RL91 was used in combination with raloxifene in in vitro studies [7,8]. Previously, we have shown that RL91 reduces Akt signaling by 98% at concentrations comparable to the ones used in this study [25]. It is worthy of note that, as PC3 cells lack phosphatase and tensin homologue (PTEN), they exhibit increased PI3K/Akt signaling [26]. Therefore, the highest synergy observed in response to raloxifene and RL91 in PC3 cells might be attributed to the dual effect on the ER and PI3K/Akt signaling pathways. These results are in agreement with the effects of raloxifene and its combination with RL91 on the cell cycle and apoptotic profile of these cell lines, as ER and PI3K/Akt signaling regulate expression of checkpoint inhibitors p21 and p27, and decrease the expression of survival proteins, such as bcl-2 [27,28,29,30]. In turn, the regulation of ER and PI3K/Akt signaling results in cell cycle arrest and apoptosis of PC3 cells. Therefore, the in vitro results suggest that raloxifene and its combination with RL91 can be useful therapeutic options in the absence of the AR. Future studies should assess the effects of this combination on PI3K/Akt and RAS/MAPK signaling.

As synergistic results were obtained in vitro, the effects of combination and individual drug treatments were examined in vivo using an orthotopic model of CRPC in mice. Orthotopic models have a number of advantages over xenograft tumors because the specific cancer microenvironment plays an important role in the growth of cancer cells. Orthotopic injections are also the most effective way to develop metastasis because it is associated with the release of viable circulating tumor cells [31]. Another advantage of using an in vivo model is to test if the drug reaches the site of the tumor [31]. Raloxifene as an individual drug was effective in the orthotopic model and reduced tumor volume by 70%. Raloxifene also reduced the mean lymph node size by 62% with no evidence of metastasis to the lungs. These tumor suppressive and anti-metastatic effects might be attributed to the reduction in ER-α and ER-β by 84 and 92%, respectively. In addition, Apoptag expression was increased by 361%. However, RL91 abolished the tumor suppressive and anti-metastatic effects of raloxifene. This phenomenon was not hypothesized; however, it has been observed previously with the combination of raloxifene and the naturally occurring compound epigallocatechin-gallate (EGCG) [17,18]. The combination was cytotoxic toward 95% of breast cancer cells when treated with raloxifene (5 µM) plus EGCG (25 µM) compared to only 30% and 50% of cells by either compound alone, respectively [18]. The reduction in cell number was associated with significant increases in apoptosis, G1 phase cell cycle arrest and decreased phosphorylated protein expression of EGFR, Akt and mTOR [17,18]. However, the synergistic effect was abolished in vivo. This effect was postulated to be due to alterations in pharmacokinetics, specifically through the increased removal of raloxifene through a possible induction of UGT1A1 caused by EGCG. This mechanism may be directly relevant to the interaction with RL91, a synthetic curcumin derivative, as curcumin has been reported to inhibit UGT catalytic activity towards naphthol while inducing UGT1A1 gene expression [32,33]. Naphthol is widely regarded as a substrate for UGT1A6, but no specific evidence is available regarding curcumin or RL91 for the UGT1A1 isoform (which metabolizes raloxifene). We hypothesize that RL91 may increase UGT1A1 expression or activity, and therefore increase the metabolism of raloxifene. This would ultimately reduce the efficacy of raloxifene in vivo. However, this theory requires experimental confirmation.

In addition to changes in UGT1A1, polymorphisms in the transporter proteins SLCO1B1 and ABCB1 have been reported to influence the response of osteoarthritis patients to raloxifene treatment [34,35]. Interestingly, alterations in the expression of the ATP-binding cassette (ABC) transporters has been linked to the activation of the PI3K/Akt/mTOR signal cascade, particularly in a PTEN negative environment [36,37]. Therefore, the inactivation of the PI3K/Akt/mTOR pathway would be expected to alter the expression of the ABC transporters which in turn would dramatically alter the in vivo efficacy of the drugs. The additional discovery that the levels of CYP3A and ABCB1 are both coordinated through the activity of the SXR orphan receptor provides further evidence that drug transport and metabolism are coordinated [38]. This strongly suggests that there may be a link between the mechanism of action of raloxifene and RL91 in terms of the EGFR/Akt pathway and changes in key pharmacokinetic parameters. Additionally, RL91 may be antagonising the effect of raloxifene and thus altering the dose of RL91 could alter this effect. However, more studies are required in order to determine the exact nature of the drug interaction.

Overall, this study adds weight to the claim that raloxifene could be a potential treatment for AR-prostate cancer. Our data shows that raloxifene had a significant growth inhibitory effect both in vitro and in vivo and also reduced lymph node metastasis. This supports previous results showing the effectiveness of raloxifene in models of prostate cancer. Our future investigations will focus on determining the exact role of ER-α and -β as well as the mechanism(s) behind the decreased efficacy caused by the combination therapy. Determining these mechanisms will be critical for the clinical use of raloxifene in order to prevent loss of efficacy due to drug interactions.

## Figures and Tables

**Figure 1 biomedicines-10-00853-f001:**
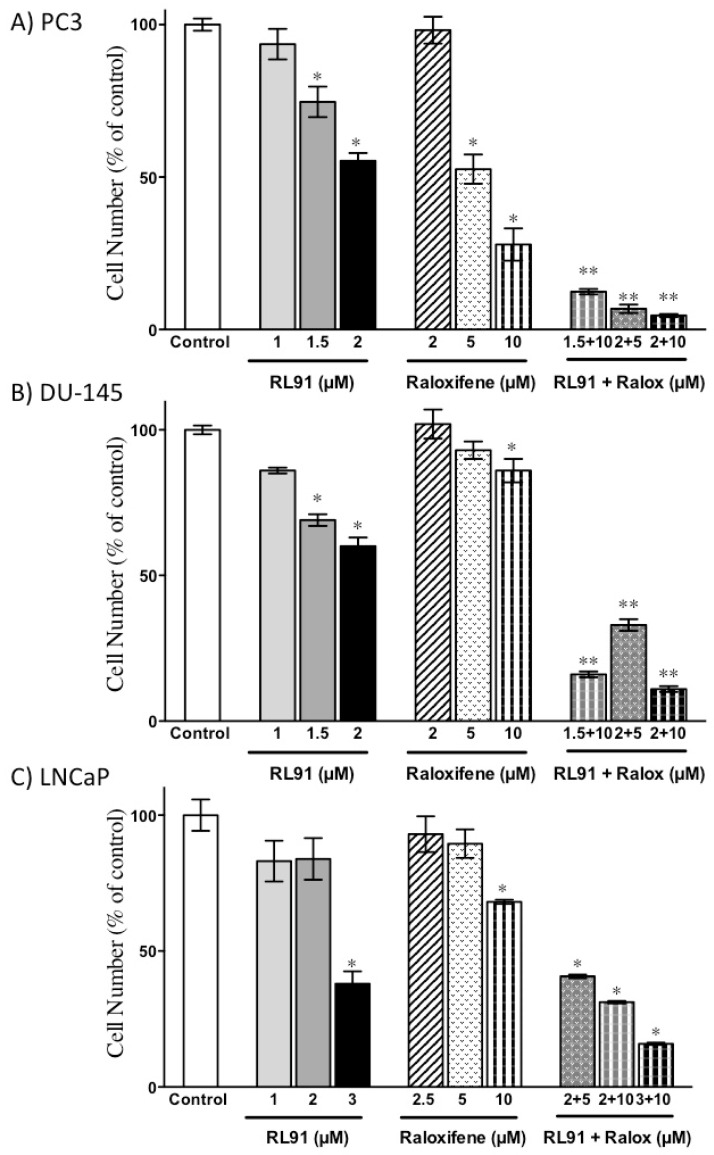
Cytotoxicity in CRPC cells following combination treatment. (**A**) PC3, (**B**) DU-145 or (**C**) LNCaP cells were treated with either RL91 (1–3 µM), raloxifene (2–10 µM), or RL91 + raloxifene for 72 h. Cell number was determined using the SRB assay. Bars represent the mean ± SEM from three independent experiments performed in triplicate. Significance was determined with a one-way ANOVA coupled with a Bonferroni post-hoc test. * Significantly decreased from control, *p* < 0.05. ** Significantly decreased from the corresponding individual treatments, *p* < 0.01.

**Figure 2 biomedicines-10-00853-f002:**
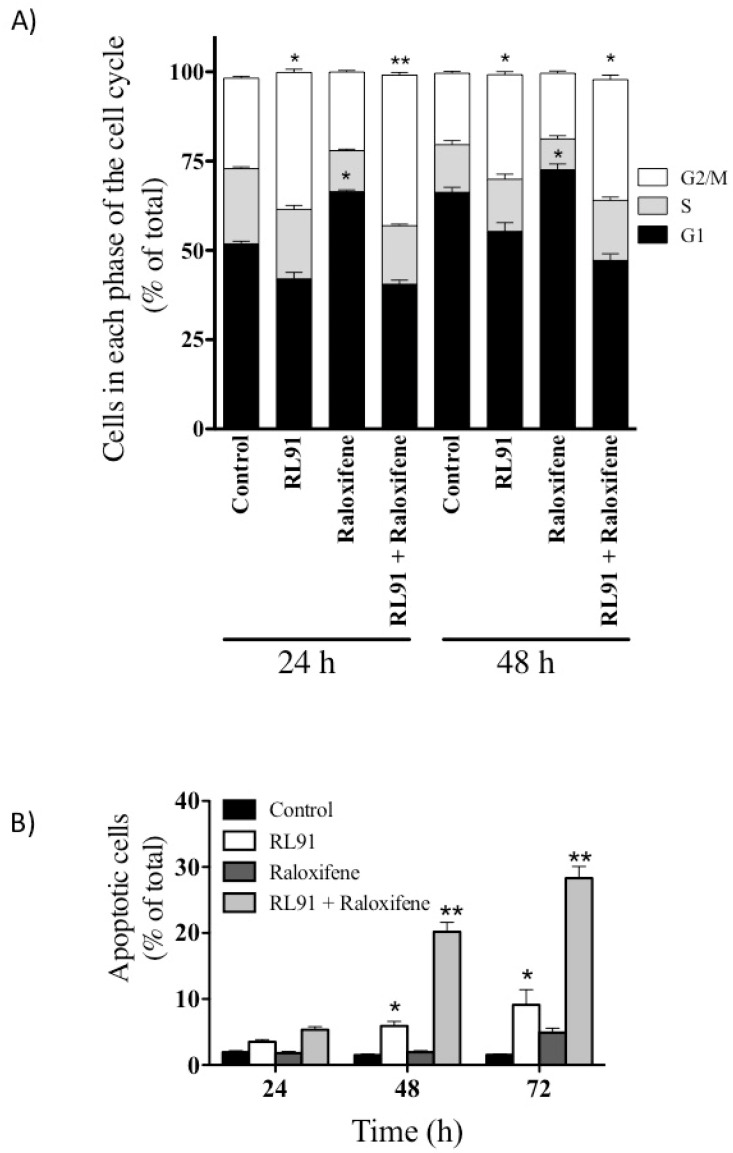
Modulation of the cell cycle and apoptosis in PC3 cells. PC3 cells were treated with either RL91 (2 µM), raloxifene (10 µM), or RL91 + raloxifne for 24–72 h. (**A**) Number of cells in each phase of the cell cycle. (**B**) Number of apoptotic cells. Bars represent the mean ± SEM from three independent experiments performed in triplicate. Statistical analysis was carried out using a two-way ANOVA with a Bonferroni post-hoc test. * Significantly decreased from control, *p* < 0.05. ** Significantly decreased from all other treatments, *p* < 0.01.

**Figure 3 biomedicines-10-00853-f003:**
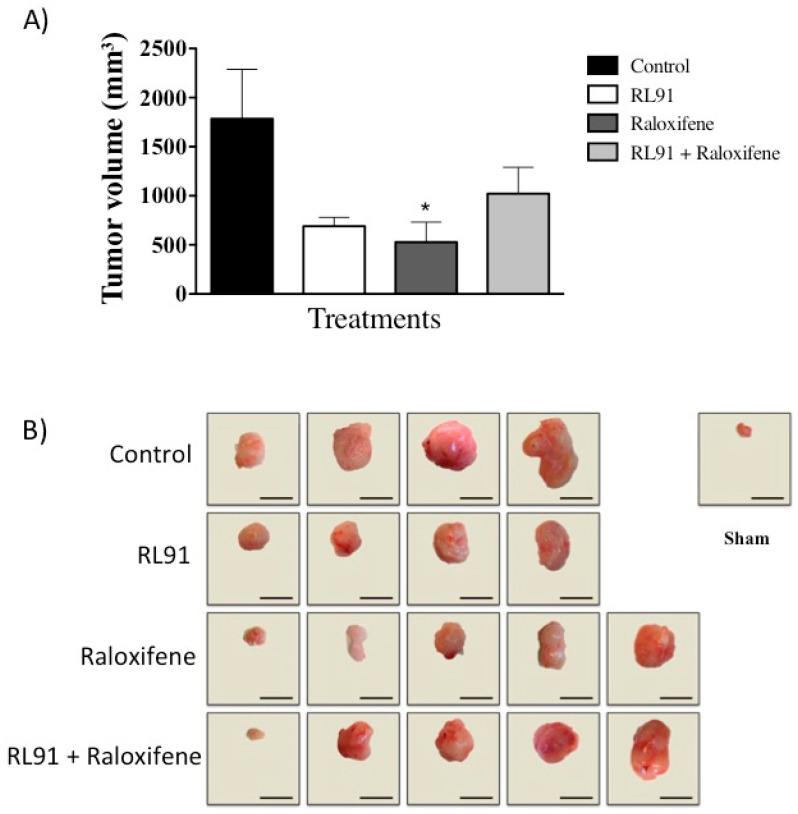
The effect of Raloxifene and RL91 on tumor growth in an orthotopic model of CRPC. Male SCID mice were inoculated with 2 × 10^5^ PC3 cells directly to the ventral prostate. Mice were treated with either vehicle (1:1 peanut oil: water 5 mL/kg), RL91 (8.5 mg/kg), raloxifene (8.5 mg/kg) or RL91 and raloxifene (8.5 mg/kg each) by oral gavage daily for six weeks. Necropsies were performed 24 h after the last day of treatment. (**A**) Mean ± SEM of tumor volume. (**B**) Photographs of the prostate tumors from each treatment group. The prostate of a sham animal that did not receive PC3 cell inoculation was included for reference. Scale bar represents 10 mm. Bars represent the mean ± SEM from *n* = four for control and RL91 and *n* = five for raloxifene and RL91 + raloxifene. Statistical analysis was carried out using a one-way ANOVA with a Bonferroni post-hoc test. * Significantly decreased from control, *p* < 0.05.

**Figure 4 biomedicines-10-00853-f004:**
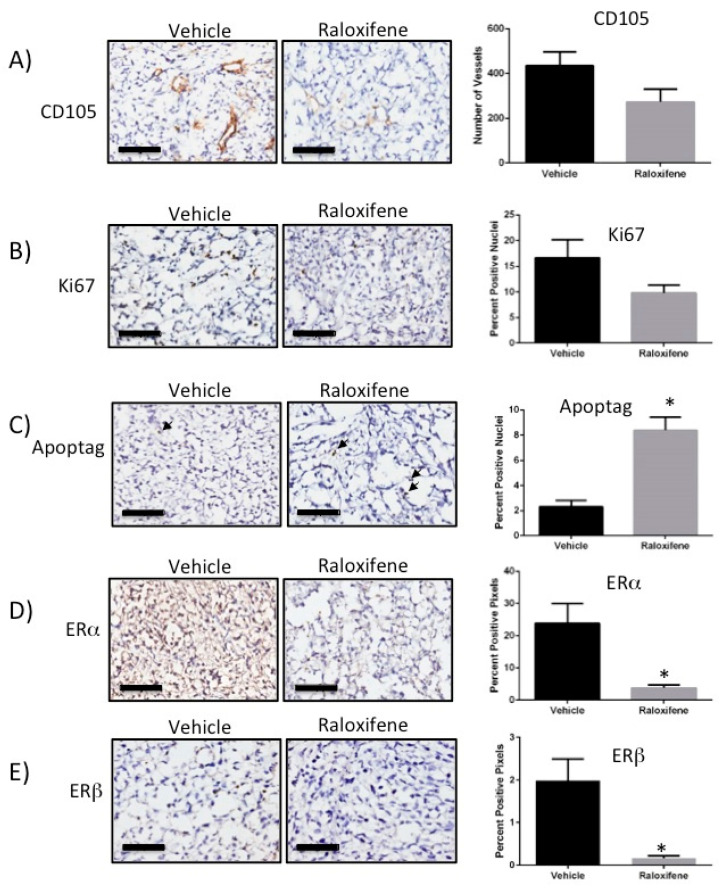
Raloxifene induces apoptosis and modulates the ER in an orthotopic model of CRPC. Following necropsy, prostate tumors from control and raloxifene treatment groups were sectioned and processed for analysis by immunohistochemistry. One representative slide is shown for each analysis. (**A**) CD105, (**B**) Ki67, (**C**) ApopTag, (**D**) ERα, (**E**) ERβ. Scale bar represents 100 µm. Bars represent the mean ± SEM from *n* = four for control and *n* = five for raloxifene. Statistical analysis was carried out using an unpaired *t*-test. * Significantly decreased from control, *p* < 0.005.

**Figure 5 biomedicines-10-00853-f005:**
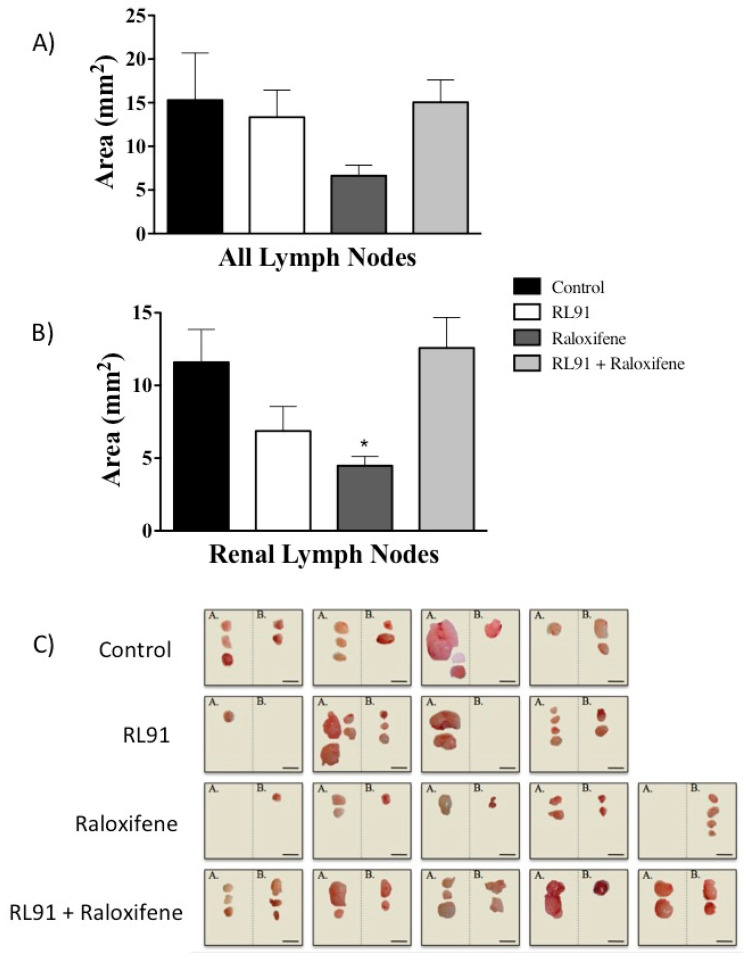
The effect of Raloxifene and RL91 on lymph node area in an orthotopic model of CRPC. Lymph nodes were removed at necropsy. Surface area was measured using ImageJ software. (**A**) Lymph node area from all lymph nodes. (**B**) Lymph node area from renal lymph nodes. (**C**) Photographs of lymph nodes removed divided into (**A**) lumbar and (**B**) renal regions. Scale bar represents 5 mm. Bars represent the mean ± SEM. Statistical analysis was carried out using a one-way ANOVA with a Bonferroni post-hoc test. * Significantly decreased from control, *p* < 0.05.

**Figure 6 biomedicines-10-00853-f006:**
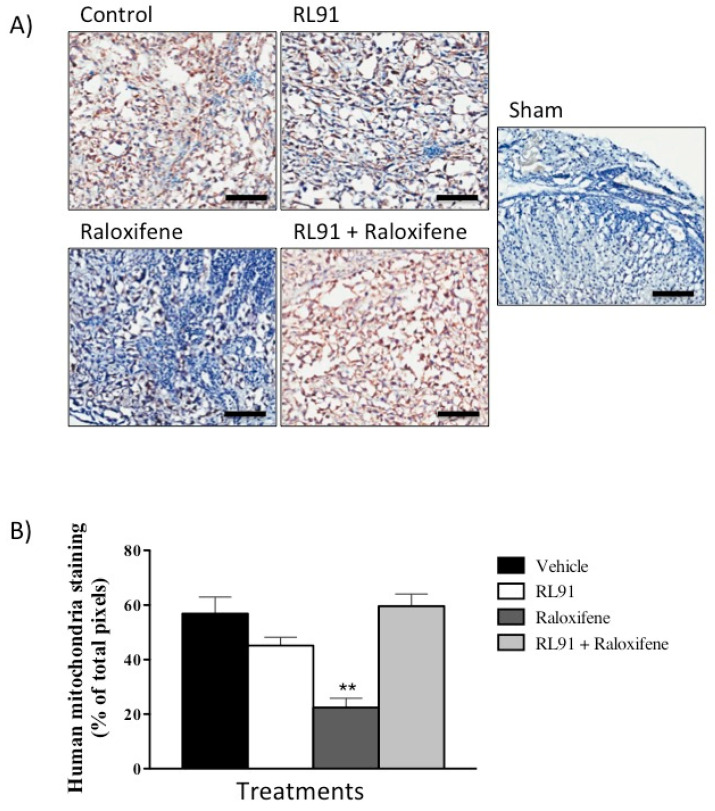
The effect of Raloxifene and RL91 on lymph node metastasis in an orthotopic model of CRPC. Lymph nodes were removed at necropsy, sliced (10 µm) and stained with human mitochondria primary antibody and hematoxylin nuclear counter stain. Brown stain represents human mitochondria and blue stain represents lymph tissue. Images were captured using Aperio and Image Scope software at 200× magnification. (**A**) One representative slide from each treatment group. Scale bar is 100 µm. (**B**) Bars represent the mean ± SEM from all lymph nodes from each treatment group. Statistical analysis was carried out using a one-way ANOVA with a Bonferroni post-hoc test. ** Significantly decreased from all other treatment groups, *p* < 0.005.

**Table 1 biomedicines-10-00853-t001:** EC50 values (µM) for raloxifene and RL91 in CRPC cell lines.

Treatments	PC3	DU-145	LNCaP
Raloxifene	11.6 ± 0.5	15.0 ± 0.6	15.0 ± 0.8
RL91	1.9 ± 0.1	2.1 ± 0.3	3.5 ± 0.2

**Table 2 biomedicines-10-00853-t002:** Indicators of animal health following six weeks of drug treatment.

Treatments	Control	Raloxifene	RL91	RL91 + Raloxifene	Sham
Body weight gain (g)	2.2 ± 0.5	2.2 ± 0.6	2.7 ± 0.4	2.9 ± 0.3	2.0 ± 0.6
ALT activity (IU/L)	31.5 ± 9.2	23.3 ± 3.8	26.2 ± 11.6	18.6 ± 6.0	29.8 ± 6.3
Creatinine (mg/dl)	0.32 ± 0.11	0.35 ± 0.06	1.25 ± 0.82	0.69 ± 0.35	0.68 ± 0.12
Liver weight (% body weight)	5.5 ± 0.11	5.1 ± 0.3	4.7 ± 0.3	4.8 ± 0.4	5.1 ± 0.2
Kidney weight (% body weight)	1.9 ± 0.05	1.9 ± 0.07	2.0 ± 1.5	2.0 ± 0.2	2.0 ± 0.04
Spleen weight (% body weight)	0.33 ± 0.06	0.24 ± 0.04	0.30 ± 0.04	0.26 ± 0.04	0.10 ± 0.01
Heart weight (% body weight)	0.78 ± 0.03	0.8 ± 0.07	0.73 ± 0.05	0.66 ± 0.04	0.80 ± 0.06
Lung weight (% body weight)	1.03 ± 0.03	0.84 ± 0.05	1.05 ± 0.13	0.86 ± 0.09	1.18 ± 0.13
Testes weight (% body weight)	0.78 ± 0.16	0.56 ± 0.02	0.78 ± 0.18	0.68 ± 0.09	0.73 ± 0.05

## Data Availability

The data presented in this study are available within the article, associated Appendix A, or on request from the corresponding author.

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
