# Peer review of "Raloxifene Suppresses Tumor Growth and Metastasis in an Orthotopic Model of Castration-Resistant Prostate Cancer"

_biomedicines, 2022, doi:10.3390/biomedicines10040853_

Round 1

Reviewer 1 Report

In this investigation Palmer et al. examined the cytotoxic potential of raloxifene and the synthetic curcumin derivative RL91 against CRPC. Their in vitro studies confirmed the therapeutic inhibition of CRPC growth by raloxifene and RL91. They showed raloxifene has tumor suppressive and anti-metastatic effects in AR negative CRPC. However, authors need to address the following concerns.

  1. Table 1 stated the EC50 values of raloxifene and RL91. Have authors calculated the IC50 values. If so, describe the variance between EC50 and IC50 values.
  2. Figure 2 depicted the growth arrest of raloxifene and RL91 in PC3 cells. What would be the fate of this AR+ve cells during the combinatorial exposure? It would be more informative if authors can include the data pertaining to both DU145 and LNCaP.
  3. Have authors assessed the molecular signaling regulations caused by the drug treatments at both gene and protein levels
  4. In in vivo studies, do mice exhibited any signs of distress and dehydration effects post drug(s) administration during the study.
  5. Why did the combination was ineffective over tumor reduction in compared to single treatments? Explain the scientific premise.
  6. Authors explained the increase of lymph nodes size after raloxifene + RL91 treatments. Did the mice showed any autoimmune disease symptoms.
  7. What is the effect of drug combination on complete blood profile post administration?
  8. The manuscript can be further revised for grammatical and typological errors.

Author Response

 #1 - Unfortunately there is different use of terminology among researchers. Technically for cell viability experiments EC50 (the effective concentration to decrease the response by 50%) is the correct terminology. However, many researchers instead express their values as IC50 (the concentration that causes inhibition by 50%). In our experiments we are not inhibiting anything, we are decreasing the number of viable cells. Therefore, we have expressed the results as EC50.

#2 - I can see why the review would be interested in cell line-specific changes in the cell cycle arrest and induction of apoptosis. However, given that we wanted to probe the mechanism for the most effective cell line and that we then used that same cell line to determine changes in orthotopic tumor growth and metastasis, we feel that this type of cell-line specificity would be better placed in a manuscript examining cell-specific effects. Given that the AR+ cells were less responsive to the drugs, there will be clear differences in apoptosis and cell cycle. Once this is shown, protein drives of this response would then need to be probed. These differences, while likely interesting, we feel are outside the scope of this manuscript that is more focused on the effects of the drugs in vivo.

3 - We have not assessed any other parameters than those outlined in the manuscript.

#4 - We have added the following sentence to the results paragraph on page 7 that describes body weight gain etc. 

Furthermore, food and water consumption as well as mouse behavior was the same in all treatment groups.

#5 - On page 13 lines 413 - 416 we hypothesised why the combination reversed the effect. Earlier in that same paragraph we back up our proposed hypothesis with detail from combination studies with EGCG and raloxifene where EGCG + raloxifene also produced the opposite effects in vitro and in vivo (i.e., EGCG reversed the activity of raloxifene.

#6 - The mice are athymic nude mice and thus they did not show any autoimmune disease symptoms.

#7 - We did not perform a complete blood profile in any of the treatment groups.

Reviewer 2 Report

The paper by Palmer and colleagues aimed to explore the effect of raloxifene on tumour phenotype in AR- PCa cells and in an orthotopic mouse model. The paper is well written, the results are clearly described, and the rationale is well documented. 

Specific comments:

1_page 3 in the paragraph 2.6 there are some space between words, also in discussion section.

2_page 10 lanes 4-6 better clarify the concept.

Author Response

point #1 - The changes in spacing are a function of justifying the margins which results in some extra spaces being inserted.

Point#2 - the paragraph has been modified for clarity and now reads as follows:

To determine whether any of the drug treatments altered metastasis of the PC3 cells, the lymph nodes and lungs from all experimental animals were examined. Similar to the results from the primary tumors, the combination treatment of RL91 and raloxifene did not alter metastasis, as determined by measuring lymph node size (Figure 5). While raloxifene decreased the lymph node size when all lymph nodes were grouped together, this effect was not significantly different from vehicle control (Figure 5A). However, when lymph nodes were analyzed by their regional location, raloxifene treatment decreased the size of renal lymph nodes 62% compared to vehicle treated mice (4.5 ± 0.6 vs 11.6 ± 2.26 mm2, respectively) (Figure 5B ).